# Wetting Characteristics of Nanosilica-Poly (acrylic acid) Transparent Anti-Fog Coatings

**DOI:** 10.3390/polym14214663

**Published:** 2022-11-01

**Authors:** Sevil Turkoglu, Jinde Zhang, Hanna Dodiuk, Samuel Kenig, Jo Ann Ratto, Joey Mead

**Affiliations:** 1Plastics Engineering Department, University of Massachusetts Lowell, Lowell, MA 01854, USA; 2Department of Polymer Materials Engineering, Shenkar College, Ramat Gan 5252626, Israel; 3U.S. Army Combat Capabilities Development Command Soldier Center, Natick, MA 21005, USA

**Keywords:** superhydrophilicity, anti-fogging, surface wetting, nanocomposite coating

## Abstract

The effect of particle loading on the wetting properties of coatings was investigated by modifying a coating formulation based on hydrophilic silica nanoparticles and poly (acrylic acid) (PAA). Water contact angle (WCA) measurements were conducted for all coatings to characterize the surface wetting properties. Wettability was improved with an increase in particle loading. The resulting coatings showed superhydrophilic (SH) behavior when the particle loading was above 53 vol. %. No new peaks were detected by attenuated total reflection (ATR-FTIR). The surface topography of the coatings was studied by atomic force microscopy (AFM) and scanning electron microscopy (SEM). The presence of hydrophilic functional groups and nano-scale roughness were found to be responsible for superhydrophilic behavior. The surface chemistry was found to be a primary factor determining the wetting properties of the coatings. Adhesion of the coatings to the substrate was tested by tape test and found to be durable. The antifogging properties of the coatings were evaluated by exposing the films under different environmental conditions. The SH coatings showed anti-fogging behavior. The transparency of the coatings was significantly improved with the increase in particle loading. The coatings showed good transparency (>85% transmission) when the particle loading was above 84 vol. %.

## 1. Introduction

Wettability has attracted great deal of attention by researchers and engineers over the past few decades [1,2,3,4,5]. Several studies have been focused on tuning the wetting behavior by manipulating the surface chemistry and roughness [6,7,8,9,10]. Superhydrophobicity [11,12,13,14,15,16] and superhydrophilicity (SH) [7,17] are the two extreme cases of wetting characteristics that can be categorized by contact angle. Superhydrophilicity is defined as an extreme water-loving surface having the static water contact angles less than 10° [18,19,20,21]. Superhydrophilic surfaces are used in many practical applications such as antifogging [22,23,24,25], antireflective [26,27], antifouling [28,29], and self-cleaning [23,30,31]. Fog formation occurs when the air is supersaturated with water vapor. A SH surface has the capacity to prevent fog formation by allowing the moisture to condense as a continuous thin film rather than forming water droplets, that restrict light transmission by scattering the light [23,27]. Highly transparent anti-fog surfaces are desired for optical lenses and windows [30]. Intensive efforts have been made to generate superhydrophilic surfaces [4,31], based on chemical surface modifications and formation of micro/nano level roughness on intrinsically hydrophilic surfaces. Surface modifications, such as plasma treatment [32], ion irradiation [33], or photo induced superhydrophilicity (PIH) [34,35], can create both changes in chemistry and roughness. Among these surface modifications, both plasma treatment and ion irradiation processes result in roughening the surface without incorporation of particles. In contrast, the PIH method requires the use of nanoparticles (NPs). Photo induced superhydrophilicity created by using light sensitive materials such as TiO_2_ was first presented by Wang et al. [36] Superhydrophilic behavior using a TiO_2_ coated glass surface is achieved upon exposing the surface to UV radiation which removes the contaminants from the hydrophobic surface and creates hydroxyl groups capable of hydrogen bonding with water. Although other inorganic particles, such as ZnO [37,38,39] and WO_3_, were reported in the literature for the PIH method, the application of PIH is limited as the light sensitive NPs are active only under UV light. In contrast, superhydrophilic surfaces based on SiO_2_ NPs do not require UV activation [24]. Hence, silica NPs are considered a promising candidate for many applications such as self-cleaning [40], anti-fogging [18] and anti-fouling [41]. In 1966, Iler found that LbL (layer by layer) coatings containing SiO_2_ NPs possess superhydrophilic or superhydrophobic properties. [42] The surface of the silica is intrinsically hydrophilic as it is covered with silanol groups (Si-OH) and water molecules [24]. Previous literature showed that the incorporation of the hydroxylated SiO_2_ NPs by direct mixing with polymers containing functional groups such as poly(acrylic acid) (PAA) and poly(ethylene glycol) (PEG) resulted in the formation of hydrogen bonding, which promotes film formation of water droplets on the surface [31]. TiO_2_/SiO_2_ NPs based multilayer coatings were found to display superhydrophilic behavior, which was attributed to the presence of nanopores in the coating [43]. In most of the coating systems, the wettability of the surfaces was improved by increasing of the weight ratio of silica NPs [31,44].

Another route for creating superhydrophilic surfaces is to introduce micro or nanoscale roughness on a hydrophilic surface [19,45]. The importance of the surface topography on wetting behavior was first indicated by Wenzel [46], followed by Cassie and Baxter [47] and more recently studied by Quere et al. [48,49] Wenzel proposed that the roughening of a surface could significantly enhance the wettability. The proposed phenomenon is expressed by the Wenzel Equation,
(1)$$cos\theta^{*}=r\;cos\theta$$
where $\theta^{*}$ and θ represent the apparent and intrinsic (Young’s) contact angles, respectively, and r is the roughness factor which represents the ratio of the actual surface area over the projected surface area. Wenzel’s theory clearly indicates that roughness can augment the wetting behavior by making hydrophilic surfaces more hydrophilic, as well as making hydrophobic surfaces more hydrophobic. Therefore, it is possible to achieve superhydrophilicity by roughening an intrinsically hydrophilic surface provided the roughness factor is sufficiently large. It has, however, also been indicated in the literature that Wenzel’s equation breaks down for highly wetting liquids or surfaces [50]. A number of processing methods have been used to make rough surfaces by depositing NPs including sol-gel [51], lithography [52], chemical vapor deposition [53] and hydrothermal treatments [54], all of which are either expensive or multi-step processes. Less complex and more simple methods to create superhydrophilic surfaces include: layer by layer assembly [22,33,56,57], dip coating [55], and spray coating [56,57].

Dong et al. presented a novel approach for preparing polymer-SiO_2_ nanocomposite coatings on glass substrates [44]. They found that hierarchical micro/nanostructure of the nanocomposite coating increased the surface roughness, decreasing the water contact angle, resulting in superhydrophilicity. Increasing the ratio of SiO_2_ to polymer decreased the water contact angle. Polakiewicz et al. presented a top down and bottom up (layer by layer) approach for preparing superhydrophilic coatings based on silica nanoparticles. [31]. Their study focused on surface topology and performance analysis of the superhydrophilic coatings. They found hydrophilicity the coatings were dependent on the combination of micro- and nano-surface roughness and the amount of silica particles. The previous works presented only a limited analysis of the role of both surface chemistry and topography.

In this work, the creation of a superhydrophilic and antifogging surface based on commercially available hydrophilic silica and a hydrophilic binder in a straight-forward process is described. The effect of particle loading on the long-term equilibrium wetting behavior of the coatings was investigated in detail. Theoretical wetting regimes (models) of the hydrophilic and superhydrophilic coatings, which have had limited studies, were extensively described and validated to explain the effect of composition on the wetting behavior of the coatings.

## 2. Experimental

### 2.1. Materials and Chemicals

Colloidal silica particles LUDOX TM-40 (40% wt. SiO_2_ suspension in water, average particle size of 22 nm, pH of 9.0 and surface area of 110–150 m^2^ g^−1^) and poly (acrylic acid) (PAA) (MW = 450,000) were purchased from Sigma-Aldrich (St. Louis, MO, USA). Plain glass microscope slides (75 × 25 mm) were used as a substrate (Fisher Scientific Company (Hampton, NH, USA) Cat. No. 12-550-A3). Deionized water was used in all rinsing processes and water-based solutions.

### 2.2. Preparation of Coatings

Schematic representation of the superhydrophilic film fabrication process is shown in Figure 1. PAA powder was dissolved in deionized water to prepare 1% wt. PAA solution by stirring (350 rpm) at 85 °C for 12 h. PAA/SiO_2_ dispersions were prepared by adding the PAA aqueous solution into a predetermined amount of hydroxylated SiO_2_ colloidal suspension (Ludox TM-40) under stirring at 350 rpm for 45 min. Coatings of PAA/SiO_2_ were prepared by dipping the bare glass slides in the different PAA/SiO_2_ suspensions having predetermined SiO_2_ concentrations. The dip-coating process was performed manually with approximately a 2 mm/s dipping and withdrawal rate. All glass slides were cleaned with isopropyl alcohol and deionized water then purged with nitrogen before coating. Coated samples were dried at room temperature for 5 min. Last, the samples were heated to 120 °C in an oven to remove any remaining solvent for 3 h then cooled to room temperature for 12h. The list of formulations is given in Table 1. The volume of the silica was calculated by taking the density of silica as 2.2 g/${\mathrm{cm}}^{3}$ which is reported in the literature [58]. The density of polyacrylic acid was measured and taken as 0.98 g/${\mathrm{cm}}^{3}$.

### 2.3. Characterization

The contact angle measurements were done using the sessile drop method (Drop Shape Analyzer–DSA100-KRÜSS GmbH, Hamburg, Germany). The measurements were performed using a 2 µL droplet volume and 2.66 µL/s dropping speed.

Scanning electron microscopy (SEM) images were taken on a field-emission scanning electron microscope (JSM 7401F, JEOL Inc., Peabody, MA, USA) typically at an electron energy of 2 to 10 kV. Coated samples were sputtered with a nanometer thin gold film to enhance the conductivity and avoid charging during scanning.

Surface topography was analyzed using an atomic force microscope (AFM) (PSIA 100) with a non-contact AFM tip. Roughness measurements were carried out by scanning a 20 × 20 μm^2^ area. Image processing and analysis was done using XEP and freely available Image J software.

Dynamic light scattering (DLS) measurements were conducted by HORIBA SZ-100 nanoparticle analyzer.

Attenuated total reflection (ATR) spectra of the samples were collected using a Fourier transform infrared (FTIR) spectrometer (Nicolet FTIR6700, Thermo Fisher Scientific Inc., Waltham, MA, USA) in reflectance mode equipped with a single-reflection diamond ATR accessory and incorporated with a KBr beam splitter and DTGS detector which operates at room temperature. Spectra were obtained in the mid- and long-wave infrared range of 4000–500 ${\mathrm{cm}}^{-1}$ and averaged over 64 scans at the spectral resolution of 4 ${\mathrm{cm}}^{-1}$. 

Tape test was done by ASTM D3359-17 standards. Lattice pattern is made by the cutting tool through the film to the substrate. Then, pressure-sensitive tape is applied on the lattice pattern and then removed carefully. The adhesion is evaluated qualitatively on a 0 to 5 scale.

The cold fog test performed by placing the substrates in humidity chamber having at least 80% humidity after keeping the substrates in a freezer (−22 °C) for 1h.

The boiling test was done by exposing the coated and bare glass substrates to the steam of the Petri glass that containing boiling water. The written letters on the paper were observed through the other side of the substrate after hot water exposure.

The transparency of the coatings was measured by UV-Vis, Hitachi U-2910 Spectrophotometer.

## 3. Results and Discussion

The wetting properties of the formulated coatings were evaluated by measuring the static water contact angle at least three times for each sample. In order to minimize the effect of the dynamic behavior on the wetting property, the measured equilibrium water contact angles (EWCA) are reported, which are the contact angle when contact lines of the water droplet stopped moving. When the prepared coatings were tested by the goniometer, the WCA measurements showed that the particle loading had a significant effect on the wetting behavior that can be clearly seen in Figure 2. An increase in particle loading resulted in a significant decrease in the water contact angle, which indicates enhanced surface hydrophilicity. Superhydrophilic surfaces having WCA lower than 10° were obtained when the vol. % of silica was 53 and higher. It was also found that the samples were able to maintain their superhydrophilic properties for at least two months.

FTIR analysis of samples is given in Figure 3. The tentative assignment of functional groups is presented in Appendix A. The FTIR spectra for other particle-loaded coatings are given in Appendix A. As seen in Appendix A, there were no discernible differences between the different loadings and the spectra were dominated by the polyacrylic acid. Because there was no common peak to use as a reference it was not possible to quantitively determine changes in functional groups of the coatings.

The microstructure of the coatings with different particle loadings was observed by FE-SEM (Figure 4 and Figure 5) It can be seen from Figure 4 that the coated surfaces are uniform and have well distributed silica particles at micro scale. In addition, the coating is composed of silica aggregates with sizes around 1 µm (particle size of the silica in the colloidal solution is around 22 nm) [59,60]. This agglomeration may be attributed to the destabilizing effect of PAA on the silica colloidal suspension system by reducing the negative surface charge on the silica NPs and changing the pH [61]. Figure 5 shows the effect of particle loading on packing. It was clearly seen that the packing was improved for high particle loading coatings. The SEM cross section images of the coatings are given in Appendix A. The thickness data of the coatings is presented in Appendix A**.**

AFM 2D and 3D images (Figure 6) revealed the topography and size distribution of silica in the coatings. The agglomerate size was obtained by analyzing the AFM images using Image J software and can be found in Table 2. According to Table 2, the agglomerate size in the dry coating decreased with increasing particle loading. As discussed above, agglomeration may be attributed to the destabilizing effect of PAA on the silica colloidal suspension system by reducing the negative surface charge on the silica NPs and changing the pH [61]. With increasing particle loading, the PAA content in suspension will be decreased and thus, the destabilizing effect will be reduced. This explains the decrease in the agglomerate size in the dry coating.

The particle size in suspension was measured by DLS (Table 3). It is clearly seen that the addition of PAA resulted in larger agglomerates in the coating suspension as compared to particles in LUDOX TM-40. The autocorrelation functions and the corresponding size distribution curves for the samples listed in Table 3 are given in Appendix A.

The roughness factor listed in Table 4 was calculated by taking the ratio of the actual surface area over geometric surface area, which were obtained from XEP software, PSIA Corp., Sungnam, Korea. The R_a_ and R_rms_ values were directly obtained from XEP software. It is interesting to note that increasing the particle loading decreased the roughness factor value, which is in contradiction to literature reports [44]. This may be related to decreased agglomerate size as particle loading increases.

Wenzel’s Equation (1) predicts improved wetting behavior (decreased water contact angle) when the surface roughness is increased for hydrophlic surfaces. However, the present data in Table 4 shows that the equilibrium water contact angle (EWCA) decreased with decreasing roughness factor values, which is in disagreement with the Wenzel theory.

To explain the results, the different wetting regimes should be analyzed. For a hydrophilic rough surface, the liquid drop may interact by two ways: by assuming Wenzel regime or film regime [48] as shown in Figure 7. The Wenzel model assumes that the space between the asperities on the surface is filled by the liquid and the drop is strongly pinned by the roughness. For the thermodynamically stable state, the intrinsic angle of the smooth surface should be larger than the critical angle (θ_c_), which is roughly determined by the roughness factor value (r). In the case of intrinsic angle being smaller than θ_c_, the roughness is impregnated and part of the liquid is imbibed [48], resulting in the water drop standing on a surface composed of solid and liquid (shown as film regime in Figure 7).

In the present case, *θ_c_* varies with particle loading and can be calculated according to Equation (2) using the roughness factor value in Table 4. The calculated *θ_c_* are listed in Table 5.
(2)$$\cos\theta_{c}=\frac{1}{r}$$

In the case of a multi-component system, which involves different species having specific intrinsic contact angles, the Cassie Equation (3) can be applied to calculate the intrinsic angle of the hybrid system where $f_{1}$ and $f_{2}$ represent the area fraction of each component [62]. In the present case, *f_1_* and *f_2_* represent silica and PAA, respectively.
(3)$$cos\theta =f_{1\;}cos\theta_{1}+f_{2}\;cos\theta_{2}$$

The intrinsic angle of a flat silica film (having no hydrophilic functional groups) has been reported as 20° [27]. It is well known that the hydrophilic functional groups such as hydroxyl and carboxyl improve the wettability by decreasing the contact angle [63]. It was also reported that the contact angle of the surfaces was reduced dramatically (more than 20 degrees) after plasma treatment from the formation of hydrophilic functional groups on the surface [63,64]. Therefore, the intrinsic angle of hydroxylated silica was assumed to be zero degrees in this work, which is 20 degrees less than a flat silica substrate. The intrinsic angle of PAA (the second species) was measured as 55° using a smooth PAA coating. Hence, the intrinsic angle for the hybrid coatings, given in Table 5, can be calculated according to Equation (3).

Comparing the calculated intrinsic angles with the critical angles in Table 5 shows that in the present system, for all particle loadings, the intrinsic angles are smaller than the critical contact angles. This indicates that the film regime is the dominant state. In this case, the EWCA should follow Equation (4) [48],
(4)$$\cos\theta^{*}=\phi_{s}cos\theta +\left(1-\phi_{s}\right)$$

Here, $\theta^{*}$ is EWCA, θ is intrinsic angle, and $\phi_{s}$ is the solid fraction which is always positive and smaller than 1. $\phi_{s}$ is mostly used to represent (%) of wetted area of the superhydrophobic coatings. However, $\phi_{s}$ in this paper refers the remaining dry area for superhydrophilic surfaces.

Thus, the relationship between cos$\theta^{*}$ and cosθ, using the values in Table 5, is shown in Figure 8. A nearly linear relationship between cos$\theta^{*}$ and cosθ is obtained which follows Equation (4). By fitting the experimental data to Equation (4), the constants are obtained. For this system the calculated value of $\phi_{s}$ is 0.16. For superhydrophobic surfaces, both the experimental and theoretical $\phi_{s}$ value is reported [15,65] and is well known. While $\phi_{s}$ 1 is expected in many cases [66], to the best of our knowledge, there are no reports of the experimental solid fraction value for superhydrophilic surfaces. Fernández-Blázquez et al. [32] correlated an area fraction of the solid–liquid interface to the top area fraction and observed that these two values were similar. In their study, the top area fraction of the plasma treated superhydrophilic (CA almost 0°) samples was reported as 12%, which is similar to the $\phi_{s}$ value obtained in this paper (0.16).

The data presented in this paper showed that the wetting behavior of the prepared samples followed the film regime model. According to the film regime model, the apparent contact angle of the nanocomposite hybrid coating is dominated by the intrinsic angle of the prepared coatings, which is determined by the surface energy of each species of the multicomponent system, rather than the surface roughness. Although superhydrophilicity requires a combination of surface roughness and chemistry, the high surface energy of the silica plays more important role leading to superhydrophilicity in our system. In principle, a further improved wettability can be expected for even higher silica content coatings. However, for the coatings with silica content (>95%) in dry state, durability and coatability could be an issue due to the lack of binder.

### The Performance Analysis of the Films

The cold fog test was performed to assess the antifog performance of the coatings [6,22]. The images in the humidity chamber for the fog evaluation are shown in Figure 9. The coated glass (b) successfully prevents fog formation, while the uncoated bare glass (a) area is fogged.

Another test conducted to assess antifog performance of the coatings was the boiling test [24,27]. The coated and uncoated glass samples placed on a Petri dish containing hot water is seen in Figure 10. The coated area prevented fog formation and allowed the letters to be easily seen by the naked eye. The uncoated area was fogged and resulted in a loss of visual clarity by restricting the light transmission.

The transparency of the coatings was measured in the visible light spectrum range (400–700 nm) which can be seen in Figure 11. The increase in particle loading sharply improved the coating transparency. The coatings having particle loading 84 vol. % and above showed good transparency (>85% transmission).

Interfacial adhesion of the coatings was analyzed by the tape test, which is widely used to study coating adhesion [67]. Adhesion is considered a critical test for coating durability and is used to determine how well the coating adheres to the substrate. ASTM D3359 defines the ratings which vary from 0B to 5B, with 5B being the highest rating for tape test durability [68,69]. In this work, the formulated coatings showed a 4B-5B adhesion level until 91 vol. % particle loading as shown in Table 6. The optical microscope images of the coatings before and after the adhesion test are given in Appendix A. The superhydrophilic coatings remained intact after the tape test and indicated proper adhesion between the coating and the substrate.

## 4. Conclusions

Superhydrophilic thin film coatings composed of silica NPs and PAA were prepared by a simple dip coating method and the relationship between the particle loading and wetting characteristics was studied. Contact-angle results showed that an increase in particle loading decreased the WCA. Superhydrophilic behavior was seen above 53 vol. % particle loading. The FTIR spectrum of the coatings did not indicate the formation of new bonds. The superhydrophilic coatings showed good interfacial adhesion, as well as consistent morphology. SEM images showed that higher particle loadings resulted in better packing. AFM analysis revealed that the coating surface was composed of nano roughness that contributes to the surface hydrophilicity. Increasing the particle loading decreased the roughness factor value. The superhydrophilicity of the coating was attributed to the presence of hydrophilic functional groups and nano-scale roughness. Based on this work, for the formulated coatings, the film regime was found to be the dominant regime describing the wetting behavior. The antifogging tests, including cold fog and boiling tests, showed that the superhydrophilic coatings successfully prevented the development of fog and can provide coatings that prevent loss of visual clarity for a variety of applications.

## Figures and Tables

**Figure 1 polymers-14-04663-f001:**
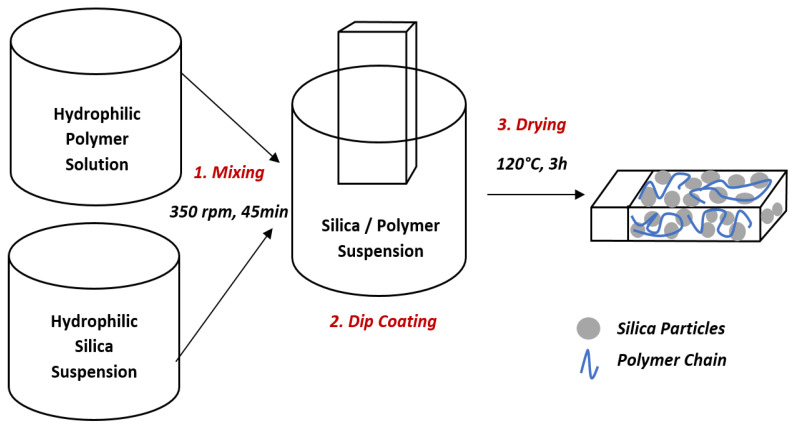
Superhydrophilic coating fabrication process.

**Figure 2 polymers-14-04663-f002:**
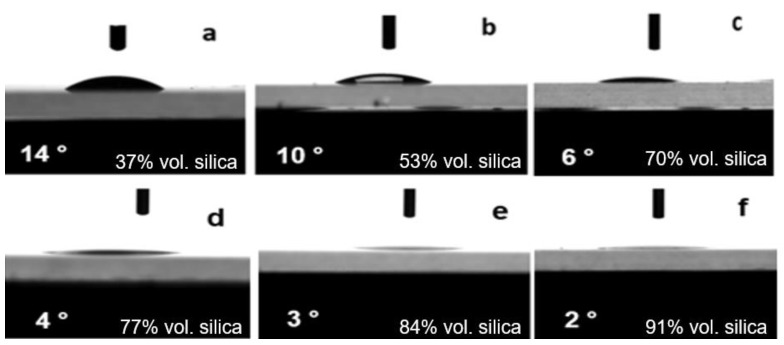
Variation of equilibrium water contact angle of coatings with particle loading (vol. %) of (**a**) 37, (**b**) 53, (**c**) 70, (**d**) 77, (**e**) 84, (**f**) 91.

**Figure 3 polymers-14-04663-f003:**
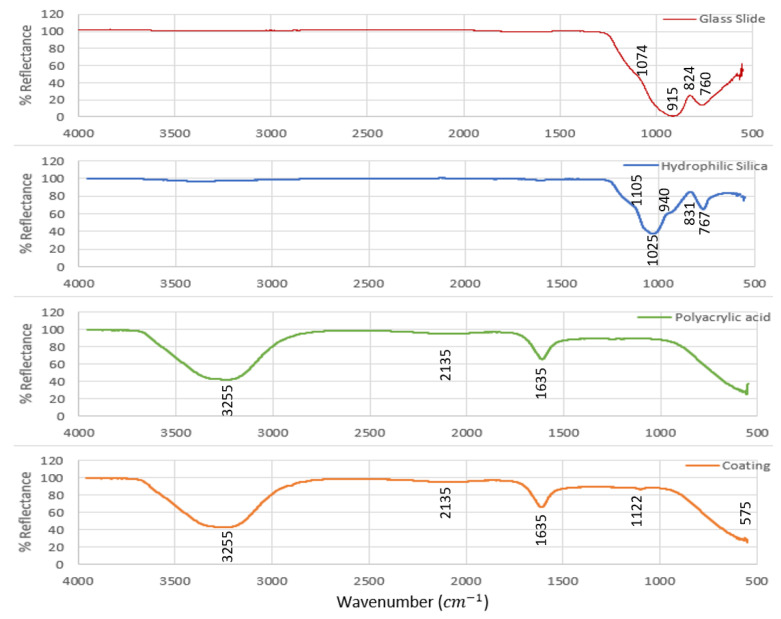
FTIR spectra of the glass slide, hydrophilic silica, polyacrylic acid and coating with particle loading 95 vol. %.

**Figure 4 polymers-14-04663-f004:**
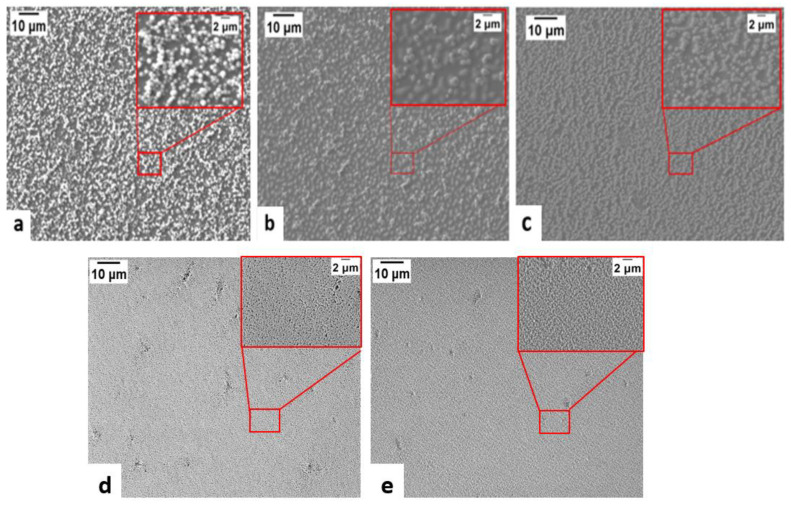
SEM images of coatings with particle loading (vol. %) of (**a**) 37, (**b**) 53, (**c**) 70, (**d**) 77, (**e**) 84. The inset is zoom-in SEM image.

**Figure 5 polymers-14-04663-f005:**
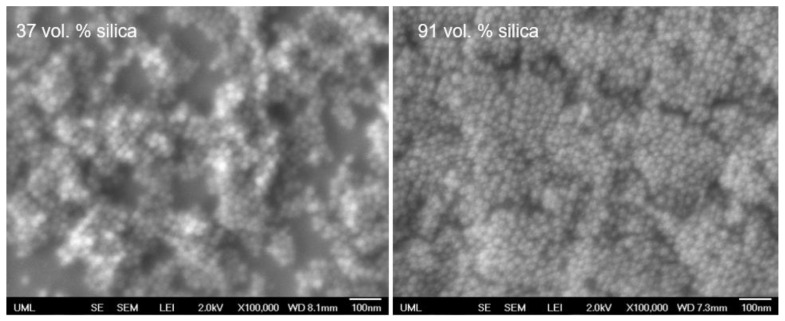
SEM image of coatings with different particle loadings.

**Figure 6 polymers-14-04663-f006:**
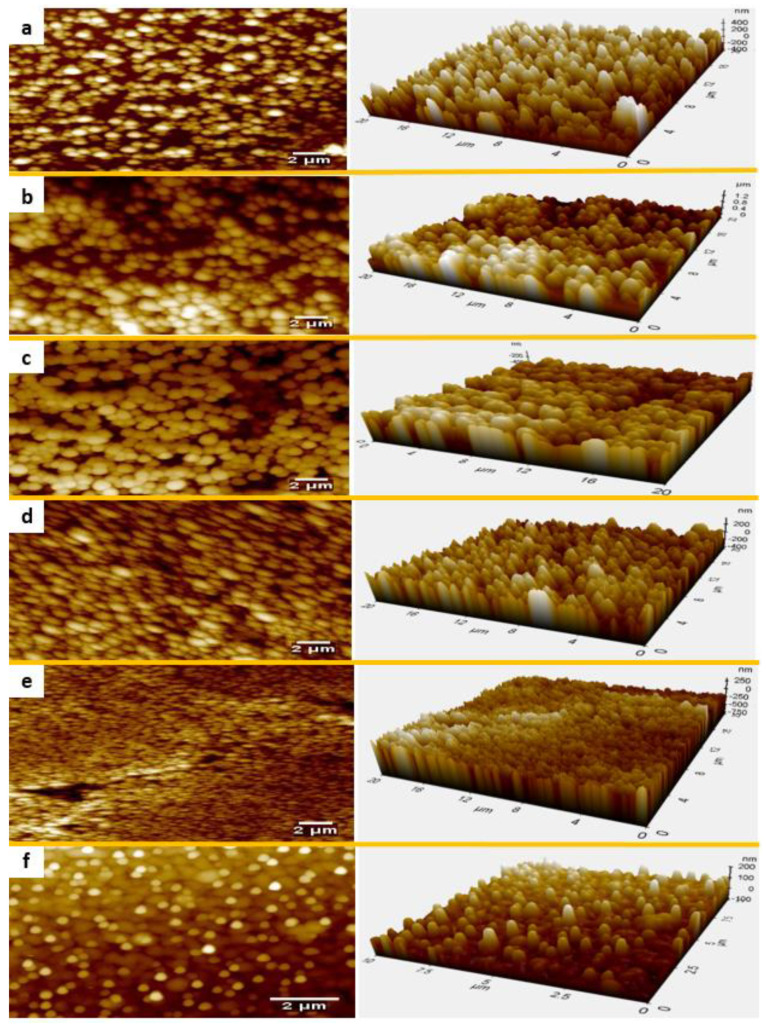
AFM topography of coatings with particle loading (vol. %) of (**a**) 37, (**b**) 53, (**c**) 70, (**d**) 77, (**e**) 84, (**f**) 91.

**Figure 7 polymers-14-04663-f007:**
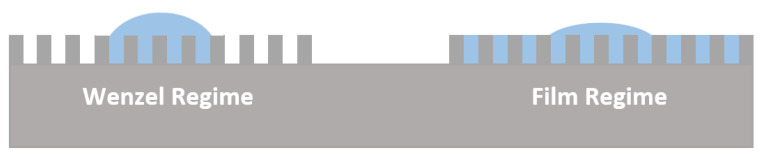
Schematic of water droplet behaviors on a rough hydrophilic surface.

**Figure 8 polymers-14-04663-f008:**
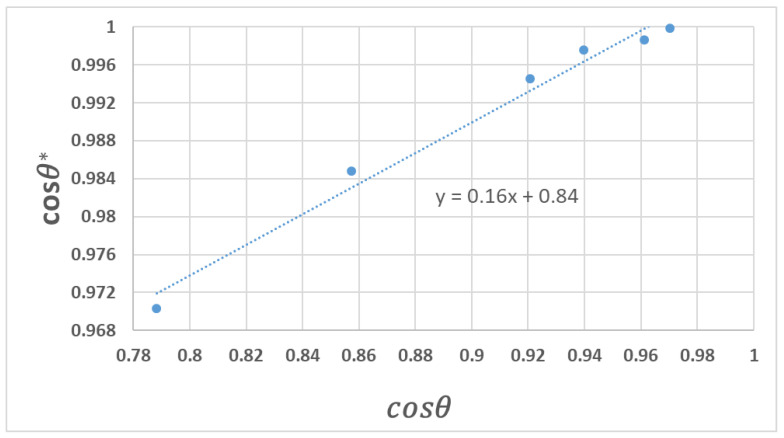
Plot of cos$\theta^{*}$ vs. cosθ.

**Figure 9 polymers-14-04663-f009:**
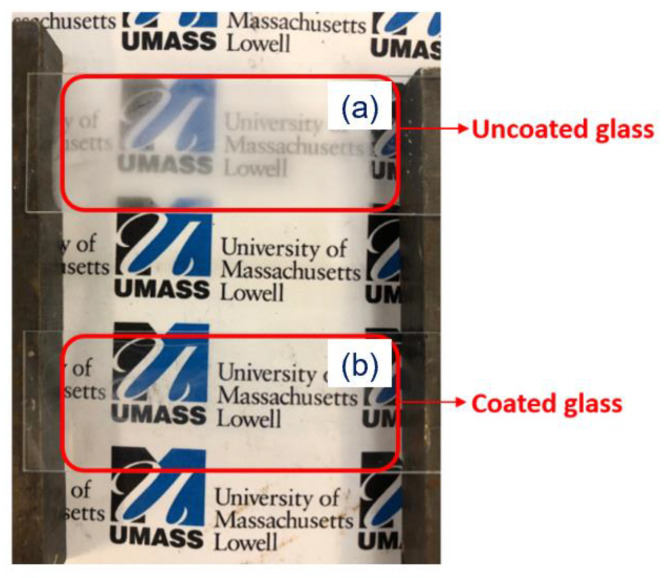
(**a**) uncoated glass, (**b**) SS6 coated glass.

**Figure 10 polymers-14-04663-f010:**
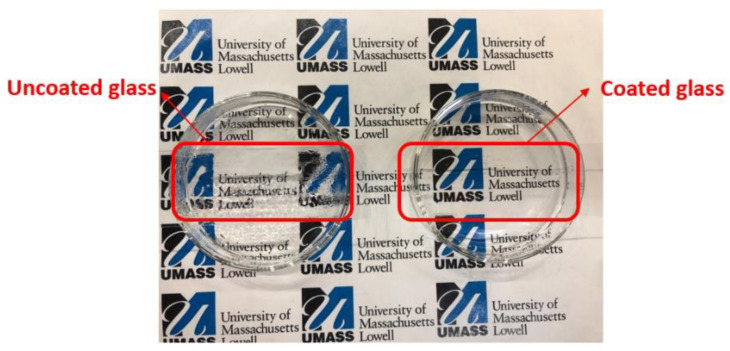
Bare glass (**left**) and SS6 coated glass (**right**) holding over a glass Petri dish containing hot water.

**Figure 11 polymers-14-04663-f011:**
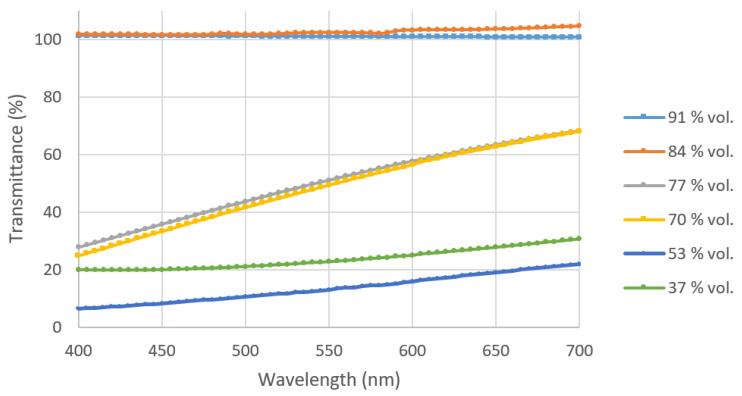
The transmittance (%) versus wavelength graphs of different coatings.

**Table 1 polymers-14-04663-t001:** Coating Formulations.

Sample Set	% wt. of Silica in Dry Coating	% vol. of Silica in Dry Coating	Particle to Binder (PB) Ratio	% wt. of Silica in Suspension
SS1	50	37	1:1	0.5
SS2	66	53	2:1	0.5
SS3	80	70	4:1	0.5
SS4	85	77	6:1	0.5
SS5	90	84	9:1	0.5
SS6	95	91	19:1	0.5

**Table 2 polymers-14-04663-t002:** Agglomerate Size of the Coatings by Image J Software.

Particle Loading (% vol.)	37	53	70	77	84	91
Agglomerate size (μm)	0.98 ± 0.17	0.97 ± 0.11	0.96 ± 0.13	0.93 ± 0.22	0.90 ± 0.11	0.60 ± 0.18

**Table 3 polymers-14-04663-t003:** The Particle Size in Suspension by DLS analysis.

Sample Name	Z-Average (nm)	Polydispersity Index (PI)
LUDOX TM-40	113	0.417
LUDOX TM-40 (5X dilution)	28	0.254
1%PAA (5X dilution)	24,830	2.990
SS6 (5X dilution)	81	0.264

**Table 4 polymers-14-04663-t004:** Statistical Parameters for Several Compositions.

Particle Loading (% vol.)	37	53	70	77	84	91
Roughness Factor (r)	1.38	1.32	1.20	1.09	1.06	1.02
Root Mean Squared Roughness (R_rms_)	196	193	152	115	111	113
Arithmetic Average Roughness (R_a_)	170	170	125	93	87	95
EWCA (°)	14	10	6	4	3	2

**Table 5 polymers-14-04663-t005:** EWCA, Intrinsic and Theoretical Critical Angles of Coatings with Different Particle Loadings.

% vol. of Silica in Dry Coating	EWCA (°)	Intrinsic Angle (°) Equation (3)	Theoretical Critical Angle (°) Equation (2)
37	14	38	44
53	10	31	47
70	6	23	30
77	4	20	27
84	3	16	26
91	2	14	14

**Table 6 polymers-14-04663-t006:** Adhesion level of the coatings by tape test.

Particle Loading (% vol.)	Adhesion
37	4B-5B
53	4B-5B
70	4B-5B
77	4B-5B
84	4B-5B
91	3B-4B

## Data Availability

The data that support the findings of this study are available from the corresponding author upon reasonable request.

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
