# Peer review of "Wetting Characteristics of Nanosilica-Poly (acrylic acid) Transparent Anti-Fog Coatings"

_polymers, 2022, doi:10.3390/polym14214663_

Round 1

Reviewer 1 Report

In this manuscript, superhydrophilic thin coatings composed of silica NPs and PAA were synthesized by a simple dip coating. The effect of particle loading on the wetting property, transparency, and adhesion was investigated. The transparent anti-fog film was achieved. This research is interesting to a broad of audience. I suggest to publish this paper after answering the following major questions.

1.      This manuscript is focused on transparent anti-fog coating. In the introduction part, different strategies of superhydrophilic surfaces were discussed. However, the transparency was not mentioned. It is necessary to introduce the transparent anti-fog surfaces and recent published superhydrophilic and superhydorphobic surfaces, such as (Polymers. 2022, 14, 2952ï¼›Applied Surface Science 527, 2020, 146733; ACS Applied Materials & Interfaces, 12, 2020, 24432). 

2.      In the experimental part, PAA solution and silica suspension were mixed. However, the concentration of PAA solution and silica suspension was missing.

3.      The length/width ratio of AFM images Figure 6 is not correct. In Figure d and e, the scanned topography shifted severely. I suggest the authors to decrease the scan rate to measure again.

4.      Table 2 shows that the decreased agglomerate size with increasing particle load to 91%. Could the authors explain this?

5.      The hydrophilic surface absorbs dirt, carbon, chemicals, etc. The hydrophilic surfaces could be failed after exposing to air for a few hours. How long can such superhydrophilic surface remain the wettability?

Reviewer 2 Report

The authors have investigated the effect of nanoparticle loading on the wetting characteristics of a nanoparticle-polymer composite coating formulation and documented that wettability improved with particle loading and that the presence of hydrophilic functional groups and nanoscale roughness were responsible for the coating’s superhydrophilic behavior and its antifogging properties. Although several studies have been performed along similar lines, the current study is an attempt to advance the understanding of wetting characteristics of superhydrophilic coatings and is suited for the audience and scope of MDPI polymers. However, before publishing the results, the authors are expected to address major concerns/questions and highlight them in the manuscript. 

After careful reading, I would like to offer my comments as follows.

Major comments:

1.     In the Introduction, the authors have provided an in-depth review of superhydrophilic coatings, including routes for preparing them and physical mechanism behind the behavior. However, it is not clear as to how the present work is different than the previous studies (to name a few – Dong et alLangmuir 201026 (19), 15567-15573 and Polakiewicz et alJ. Adhes. Sci. Technol. 201428 (5), 466-478) that have used nanoparticle-loaded polymer composites as superhydrophilic coatings. In the current scenario, the novelty of the work is diluted and appears repetitive.

To make the distinction clear, it would be helpful to highlight the novelty using the proposed layout: (a) key highlights of the previous studies, (b) crucial points lacking in those studies, (c) importance of the missing pieces from a big-picture perspective, (d) how does the current study help to address those to advance the understanding, and (e) how does the current findings help the field – future scope and applications of the observations. This will help the reader to appreciate the current study and improve the impact of the work.

Additionally, for the comment made in lines 86-88 (specifically 87; “not many studies have focused on”), the authors are expected to highlight (or cite) what studies have focused on using theoretical models to explain wetting/superhydrophilic regimes of coatings. In its present state, the authors have loosely furnished a statement.

2.     From the perspective of reproducibility of experiments and analyses, the authors need to expound on some method details. 

a)     How was the dip-coating process performed? What were the specifics of dipping and withdrawal rates of the glass sides into the coating mixture? Is there any dependence of those rates on the thickness of the coating deposition?

b)    How was the AFM analysis performed to obtain roughness parameters? How were the AFM images processed? What software was used to perform the image processing and analysis? How was the roughness factor (r) obtained in Table 4? Please provide details of the procedure for reproducibility.

3.     In Figure 3, it would be helpful to provide the FTIR spectra for other particle-loaded coatings, in addition to 95 vol.% case. The current system is largely dominated by the peaks of silica. It would be interesting to see if there’s a gradual decrease in the contributions of polyacrylic acid components as the particle loading increases. This would serve as a complementary dataset to the SEM images to comment on the constitution of the varying particle-loaded coatings. 

Furthermore, can the authors comment on how silica nanoparticles are aiding in wetting (chemistry wise), when the FTIR spectra show that OH group and hydrogen bonding contributions are majorly coming from the polyacrylic acid component, which is significantly small for the 95 vol.% particle-loading case? Please clearly highlight which peak assignments are responsible for aiding the wetting behavior.

4.     In Figure 4, it appears that 53 vol.% (b) and 70 vol.% (c) have more polymeric component than the 37 vol.% (a) case. Can the authors comment on this? Additionally, have the authors looked at the thickness of each coating type? Since 77 vol.% (d) and 84 vol.% (e) are densely packed films, the thickness for those should be greater than, say, 37 vol.% (a) coating. This can also be clearly observed in the SEM images in Figure 5. Does coating thickness affect wetting behavior and wetting regime?

For coatings, like in 37 vol.% (a) case, it is challenging to exclude the substrate effect on the wetting properties. Have the authors ensured that their observations are not biased by any substrate effect? If so, how have they validated this? Please include concerned details in the main manuscript as well for clarity.

5.     Following on point 4, since there appears to be two types of coatings (based on SEM images in Figure 4 and 5) – with and without exposed substrate, is it still logical to consider the system to fall in the film regime? Can the authors comment if the film regime model excludes/includes substrate effect? 

The authors should provide optical micrographs of the coated samples and/or cross-section SEM images of their coatings to comment on coating uniformity and thickness.

6.     From reference # 64 (Lai et alAppl. Surf. Sci. 2006252, 3375-3379), it is not clear as to why the authors made a rational decision to choose zero-degree intrinsic angle for hydroxylated silica. Can the authors supply a more relevant reference to support their assumption? Like for the case of PAA coating, the authors could have attempted a contact angle measurement on plasma-treated silica substrates. 

7.     The term fs in the references 15 (Zheng et alACS Appl. Polym. Mater. 20202 (4), 1614-1622) and 65 (Dorrer et alAdv. Mater. 200820, 159-163) is defined as the solid-liquid interface ratio to the entire surface area under consideration. This depicts the portion (%) of wetted area for the superhydrophobic coatings (see reference 15 for pictorial representation). Can the authors clarify the definition of fs term in the present study? If the same terminology is used, the calculated value for  fs term (0.16 in this study) is low for superhydrophilic coatings. 

8.     Line 245 – 246: The authors conclude that the primary reason leading to superhydrophilicity is the high surface energy of silica. What is the basis for arriving at this conclusion? It is not clear in the manuscript and appears abruptly. Please clarify and provide appropriate reasons, directing to data supporting this conclusion.

Minor comments:

1.     It would be helpful if the authors can provide a reference to the sentence in line 50-51: “In contrast, superhydrophilic surfaces based on SiO2 NPs do not require UV activation.”

2.     In Figure 3, there are several peaks in the FTIR spectra which have not been described by the authors in the main body of the manuscript. For the sake of readers’ convenience, it would be helpful to tabulate peak assignments. This table could be furnished in the Supporting Information.

3.     In Table 2, it would be useful if the authors can provide standard deviations for the agglomerate size (mm) of the coatings. This would help to identify if the differences between different coatings are significant or not. If possible, statistics (multiple pairwise comparison of means like Tukey HSD Test) could be used to support the conclusion of the result table.

4.     The authors should provide the autocorrelation functions and the corresponding size distribution curves for the different samples mentioned in Table 3 in the Supporting Information, if possible.

5.     For readers’ ease, it would be helpful if the authors can provide a schematic representation of the adhesion test and photographic snapshots of the outcome of the test on different coating samples. This data could be supplemented as Supporting Information.

Round 2

Reviewer 2 Report

The edits/incorporations made by the authors are acceptable to move forward with the publication. I thank the authors for their sincere effort.